# Rewilding Apex Predators Has Effects on Lower Trophic Levels: Cheetahs and Ungulates in a Woodland Savanna

**DOI:** 10.3390/ani12243532

**Published:** 2022-12-14

**Authors:** Dallas B. Ruble, Stijn Verschueren, Bogdan Cristescu, Laurie L. Marker

**Affiliations:** Cheetah Conservation Fund, Otjiwarongo 12001, Namibia

**Keywords:** *Acinonyx jubatus*, carnivore reintroduction, ecology of fear, ecosystem restoration, predation risk, trophic rewilding

## Abstract

**Simple Summary:**

We investigated the behavioral responses of five African ungulates to cheetah reintroduction in a semi-arid system affected by bush encroachment. Visitation rates, duration of stay, and activity patterns of ungulates at waterholes were compared with and without cheetah presence. During cheetah presence, visits to waterholes were rare for medium-sized species, but frequent for large-sized species, and visits were longer for small- and large-sized ungulates, possibly indicating increased vigilance. No differences in daily or seasonal activity at waterholes were found, which may be due to permanent and therefore predictable water availability. Further research into long-term behavioral consequences of trophic rewilding is recommended to maximize the success of recovery programs and minimize negative effects.

**Abstract:**

The restoration of ecosystems through trophic rewilding has become increasingly common worldwide, but the effects on predator–prey and ecosystem dynamics remain poorly understood. For example, predation pressure may impose spatiotemporal behavioural adjustments in prey individuals, affecting herbivory and predation success, and therefore potentially impinging on the long-term success of trophic rewilding through apex predator reintroduction. Predation risk might have detrimental effects on prey through displacement from water or other vital resources. We investigated how five species of African ungulates responded behaviourally to changes in predation risk, following cheetah releases in the system. We grouped ungulates by body size to represent preferred prey weight ranges of the cheetah and examined changes in visitation rates, duration of stay, and activity patterns at waterholes with and without cheetah presence. During cheetah presence, visitation rates of ungulates were low for medium-sized species but high for large-sized species, suggesting that the species within the cheetah’s preferred prey weight range adjusted behaviourally to minimize waterhole visits. Visits to waterholes were longer for small- and large-sized ungulates with cheetah presence, possibly indicating increased vigilance, or a strategy to maximize water intake per visit while minimizing visits. We did not detect significant differences in circadian or seasonal activity in waterhole visits, which may be attributable to the need of ungulates to access water year-round in our semi-arid study system and where migration was impeded due to physical barriers (fencing). We recommend further research into the long-term behavioural consequences of trophic rewilding on prey populations and trophic cascades to assist the success of recovery programs and to minimize potential detrimental effects at target sites.

## 1. Introduction

Apex carnivores structure ecosystems through various processes along multiple food-web pathways [1]. Lethal effects of predation may directly determine prey numbers, regulate disease dynamics, and facilitate competition [2,3]. Meanwhile, non-consumptive effects, such as presence of carnivores, can modulate prey behaviour to decrease predation risk by re-distributing spatially in a “landscape of fear” [4,5,6]. Declines of both prey and predator populations can destabilize predator–prey processes [7], alter vegetation communities [8,9,10] and ultimately affect ecosystem functioning [11,12]. With many systems having lost their native apex predators due to human impacts [13], trophic restoration through large carnivore reintroductions is a widely adopted strategy aiming to re-establish functional top-down interactions [14,15]. 

The reestablishment of gray wolves (*Canis lupus*) in Yellowstone is known for its impact on top-down trophic rewilding of the ecosystem [16,17,18,19]. Trophic downgrading of the Greater Yellowstone ecosystem through wolf extirpation initiated a cascade that extended from herbivores to vegetation [18,20]. During the 70-year wolf absence period, herbivory significantly increased, making it difficult for plants such as aspen and cottonwood to successfully establish [21,22]. Long-term effects of wolf reintroduction counteracted these unfavourable dynamics by reinstating herbivore behavioural processes associated with the landscape of fear in a top-down structure [4,16,18,19,22,23].

Beyond the wolves in Yellowstone, reintroductions of predators more generally contribute to ecosystem restoration. For example, sea otters (*Enhydra lutris*) restore kelp forests by foraging on sea urchins [13]. Additionally, a combination of plant anti-herbivore defenses and modified herbivore foraging patterns in response to predation risk shape the distribution of thorny Acacia trees (*A. etbaica*) [24]. Furthermore, dingoes *(Canis dingo)* shift the structure of their tropical–savanna ecosystem through control of herbivores and mesopredators [25]. 

There remains, however, a geographical bias in the limited studies that evaluate ecosystem-level effects of trophic rewilding, with most focusing on North America, Europe, and oceanic islands due to the eradication of carnivores in those areas [14,26]. Southern Africa, on the other hand, has prominent large carnivore guilds and is increasingly becoming the focus of trophic rewilding efforts through carnivore reintroductions [27,28,29,30,31,32,33], but associated top-down ecosystem-level effects are not well-studied [34]. Prey responses to changing predation pressures in savanna ecosystems can be complex and diversified [35,36], which could affect prey population dynamics as well as the rehabilitated carnivores that depend on prey for long-term viability. 

The complexity of Africa’s terrestrial mammal community results in intricate predator–prey interactions [26,37]. Prey species display different behaviours to reduce predation risk with the reward of foraging, such as increased vigilance, avoidance of high-risk areas, and temporal fluctuations in activity patterns [38,39,40]. The temporal and spatial avoidance of areas perceived as high risk can affect prey physiology and demography [41,42], plant biomass [39], and nutrient deposition [11,43]. In East Africa, predator presence elicits prey risk-avoidance behaviour which fundamentally contributes to spatial variation in vegetation [24]. The strength of behavioural responses prompted by predator presence correlates to the risk level present in the ecosystem [3,44], which is determined among other factors by predator hunting mode and body size [38,40,45,46], and is also affected by seasonal resource availability [38,39]. During droughts, waterholes may attract predators and hold higher predation risk, due to aggregation of prey around the scarce but critical resource [38,47]. Therefore, anti-predator mitigation tactics are likely sensitive to a multitude of factors that may vary by ecosystem, seasonality, and species involved. 

To better understand the ecological effects of predation pressure on the behavioural response of prey species when accessing key resources, we focused on ungulates at waterholes in a woodland savanna to determine potential changes in (i) visitation rate as an indicator of relative abundance, (ii) duration of visit as an indicator of vigilance behaviour, and (iii) activity pattern as an indicator of temporal segregation, with and without cheetahs (*Acinonyx jubatus*) experimentally released in the enclosed study system (a fenced game reserve). We expected that predator presence would result in (i) a lower frequency of ungulate visits to waterholes because of increased predation risk, (ii) a longer duration of stays at waterholes because of increased vigilance or as a strategy to maximize water intake, and (iii) a shift in activity to diurnal patterns compared to predator absence to avoid predators in time. We discuss our findings considering recent studies that report trophic cascades which influence ecological processes from the top down [9,48]. 

## 2. Materials and Methods

### 2.1. Study Area

We conducted the study in a semi-arid woodland savanna (Figure 1). Seasons in the region are wet–hot (January–April), dry–cold (May–August) and intermediate (September–December) [49]. Mean annual rainfall is 468 mm, mostly falling in the wet season [50]. The study area, Bellebenno A (40 km^2^), is a game reserve enclosed along its perimeter by ungulate-proof fencing (2.4 m height, 10 strands of wire). Ungulate densities were relatively high with minimal human influence, as no ecotourism or hunting activities occurred. Four artificial permanent waterholes (3 m × 3 m × 0.3 m) were present, with one ground-based observer lookout located approximately 30 m away from each waterhole. Common ungulate species present included common duiker *(Sylvicapra grimmia)*, greater kudu *(Tragelaphus strepsiceros)*, impala *(Aepyceros melampus)*, springbok *(Antidorcas marsupialis)*, and steenbok *(Raphicerus campestris)*. For the study duration, larger carnivores, including leopard (*Panthera pardus)* and brown hyena *(Hyaena brunnea)* were present incidentally at low densities. 

### 2.2. Ungulate Data

Between 2008 and 2019, every 2 months we conducted 12 h (06:00–18:00) direct observation surveys at the four waterholes. Two observers per waterhole positioned in the lookout recorded the following information for each animal visiting the waterhole: ungulate species, group composition (number of individuals, sex and age class), activity patterns (times of arrival and departure), and weather conditions.

### 2.3. Cheetah Releases

Between 2004 and 2014, we temporarily released wild-born, captive-raised adult cheetahs in the study area as part of developing a cheetah rehabilitation protocol. The objective of the releases was to assess whether these individuals had the skills to survive independently in the wild for future rewilding at final destination sites [33]. Following release at Bellebenno A, cheetahs moved freely within the fenced boundaries, and once deemed to be rehabilitated, they were captured and translocated to the reintroduction sites. We analyse herein the prey behavioural responses to cheetah released during 2008–2014 (hereafter, ‘cheetah presence’ period) because our waterhole surveys were started in 2008 (Appendix A). No cheetahs were present in the study area between 2015 and 2019 (hereafter, ‘cheetah absence’ period). 

### 2.4. Data Analysis

We tested for the effect of cheetah presence on the behaviour of preferred prey species of cheetahs [51,52], which we classified into 3 groups according to body size: small (duiker and steenbok), medium (impala and springbok), and large (kudu). Species with different body sizes may show different responses to predation risk [53]. Due to weather, time, and vehicular constraints, we included ungulate direct observation data from one sampling session per season. We selected survey data for March (wet–hot season), May (dry–cold season) and November (intermediate season), which were the months sampled consistently throughout the study duration. The dataset thereby comprised 36 days (12 h/day) of direct observations per waterhole, for a total of 144 waterhole sampling sessions, which summed to 1728 h.

We used a general linear model (GLM) framework to analyse the data towards our hypotheses. 

I.*Visitation rate (relative abundance) hypothesis*: We constructed two model sets that differed in the response variable. One set included independent species-specific visits of ungulates recorded per waterhole survey session, whereas the other considered herd size as the response variable, respectively. We included experimental treatment (cheetah presence/absence), ungulate body size (3 classes), and season (3 seasons) as explanatory variables, with an interaction effect between treatment and body size. To test for zero inflation, we compared a Poisson model for count data to a negative binomial model and a zero-inflated Poisson model, ranking them based on their BIC values with the R package *performance* [54]. We tested the influence of including waterhole identity as random effect, but excluded it from the model sets, as analysis of variance tests showed that the random effect did not outperform fixed effects models (Appendix A). Models were run with the function “*glmmTMB*” from R package *glmmTMB* [55] and “*zi*” from R package *pscl* [56]. Model diagnostics were evaluated with R package *DHARMa* [57].II.*Vigilance and/or resource maximization hypothesis*: We built one model with the ungulate duration of visits at the waterholes (seconds) as the response variable, and treatment, body size, and season as explanatory variables. We included an interaction effect between treatment and body size. The response variable was log-transformed to meet model assumptions of normality and homoscedasticity. Waterhole identity was again tested as a random effect, but was not retained in the final model set, as it did not explain additional variation.III.*Temporal segregation hypothesis*: We determined whether ungulates adjusted their activity patterns according to cheetah presence vs. absence using the R package *overlap* [58]. To test whether activity patterns of ungulates differed significantly with and without cheetahs in the system, we used the Watson–Wheeler test in the R package *circular* [59]. Due to sample size limitations, we did not generate activity patterns by ungulate body size class and season.

Statistical analyses were performed using R Studio version 4.1.0 [60]. Data were sorted and manipulated using R packages *tidyverse* [61] and *lubridate* [62].

## 3. Results

Over the entire sampling period, we recorded a total of 173 independent visits by ungulates within the preferred prey range of cheetahs (Table 1). The behaviour of prey species varied according to body size based on whether cheetahs were present or absent (Appendix A). The visitation rate of medium-sized ungulates was lower compared to large ungulates when cheetahs were present (Table 2). Prey species spent significantly longer periods at waterholes during cheetah presence compared to cheetah absence, but medium-sized species were excluded from this analysis due to lack of observations with cheetah presence (Table 2, Appendix A). Seasonality had no significant influence in any of the analyses. 

The activity patterns of ungulates during both treatment periods were similar (W = 4.10, df = 2, *p* = 0.13) and had relatively high overlap (Dhat1 = 0.82). Nonetheless, activity peaks appeared to show a slight shift from early morning to evening during cheetah presence (Appendix A).

## 4. Discussion

The effect of cheetah releases on the prey’s visitation rates (relative abundances) at waterholes followed our prediction for the ungulates within the preferred weight range of cheetahs (23–56 kg, i.e., medium-sized species; [51]). We documented a lower visitation rate by medium-sized species at waterholes with cheetah presence, which suggests that predation pressure by cheetahs may have a significant effect on their spatial distribution and/or demographics. Our data also possibly indicate that medium-sized prey species had a higher frequency of visits to waterholes after cheetahs were translocated from the study area. This suggests that predator releases should be carefully planned and monitored to evaluate potential secondary effects on prey. The relatively high abundance of large-sized species during cheetah presence is unlike prior studies, which found abundance and/or visitation rates of prey to decrease with carnivore presence [38]. The difference in size among prey species will result in varying risk-avoidance behaviour in response to predation risk by a given predator species [40,43], in this case the cheetah. In a recent study, small, medium, and large African ungulates were presented with lion (*Panthera leo*), African wild dog (*Lycaon pictus*), and cheetah vocalizations to represent a multi-predator system [63]. The prey under consideration exhibited the strongest fear response to the predator most likely to kill the respective prey species, and the response was related to body size and sociality of the respective predator [63]. In our study, the increase of kudu with cheetah presence could be attributed to lower predation risk for kudu from cheetah as compared to smaller herbivores available in the study system. 

Our vigilance hypothesis was supported, as the duration of ungulate visits to waterholes was higher with cheetah presence. This is consistent with Valeix et al. (2009) [38], who reported that prolonged visitation of waterholes was associated with increased vigilance and early detection of predators. Our dataset did not include ethograms of ungulate behaviour to directly quantify vigilance. Therefore, extensive duration of waterhole visits may be alternatively attributed to increased resource (water) intake when risking waterhole visitation, coupled with a decreased number of visits to minimize risk of predation [16,35]. It may as well be that the duration of ungulate visits to waterholes is related to ungulate density and herd size. Individuals in larger herds may need to alternate turns or wait for access to the waterhole. As the relative abundance of large ungulates we observed at waterholes increased with cheetah presence, this may partially explain the longer duration of visits to waterholes by large ungulates when cheetahs were present. We recommend focused research on prey behavioural states at waterholes for improved understanding of risk–reward trade-offs in ungulates when accessing critical resources.

Our temporal segregation hypothesis was not statistically supported, yet we detected a possible slight shift in prey activity peaks from morning (~6:00) to evening (~18:00) with cheetah presence. The activity patterns of ungulates in our system could be partially related to risk avoidance tactics, as cheetah activity tends to peak in the early morning (~6:00) in the broader region [64]. Studies evaluating risk avoidance behaviour by ungulates have observed stronger adaptive responses of prey species to ambush hunters (i.e., leopard) compared to cursorial hunters (i.e., cheetahs) [5,40,46,63]. 

Prey populations are regulated by a variety of factors in addition to predation pressure. These include, amongst others, the availability of resources, disease outbreaks and environmental conditions such as weather and climate [65]. It is therefore possible that fluctuations in prey density over the 12-year study duration affected our findings on waterhole use by ungulates. Although we do not have data available to assess population trends, prey populations in our enclosed study system were not anthropogenically harvested, and there was no evident die-off nor extreme environmental conditions recorded; therefore; we assumed that they were close to carrying capacity and with limited fluctuation in density.

We acknowledge sample size limitations, specifically for medium-sized ungulates and nocturnal species. The low number of observations of medium-sized ungulates may be attributed to different habitat preferences of these species, as impalas prefer open woodlands with nutrient-rich vegetation [66,67], while springbok thrive in dry, open grasslands [68]. Our study was conducted in a wooded and relatively bush-encroached system, which may present suboptimal habitat conditions for these species. Due to safety and logistical reasons, we did not record night-time activity data through our direct observation method; hence, observations of predominantly nocturnal species such as duiker were low. Camera traps are well-suited for collecting activity budget information on a 24 h basis [38,40], but were not available for the 12-year study duration, and we recommend their use in future studies that aim to determine behaviour of ungulates at waterholes.

## 5. Conclusions

Our results provide incipient evidence in a comparative framework that large carnivore presence can structure prey behaviour around vital resources, with the strongest signal for ungulates in the carnivore’s preferred prey size class (medium-bodied ungulates in response to cheetah presence in this case). Unlike other studies [38,39], seasonality did not affect waterhole use by herbivores, possibly because in our semi-arid savanna ecosystem ungulates benefit from water intake year-round. We emphasize the importance of further exploring prey behavioural adjustments when accessing critical resources, as well as investigating additional ecosystem-level effects associated with fluctuating predator pressure. The timing is right due to the increasing effort and interest in rewilding as a restoration tool [15].

## Figures and Tables

**Figure 1 animals-12-03532-f001:**
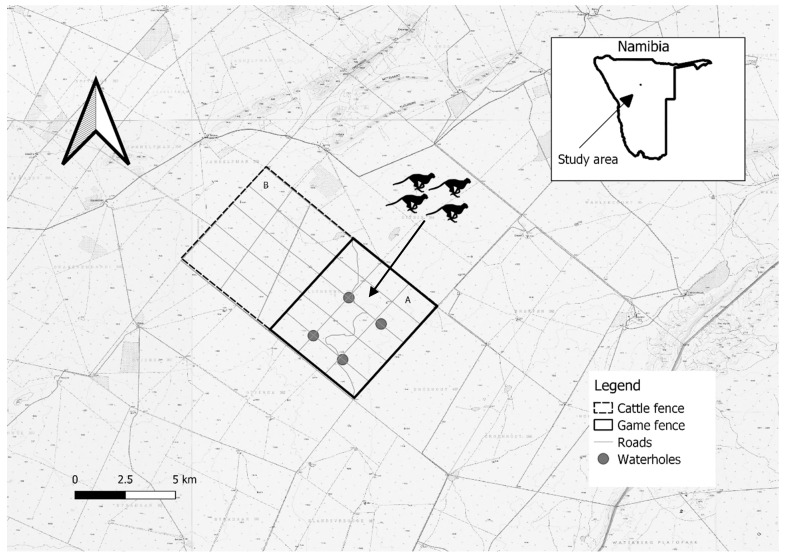
Map of Bellebenno game reserve, Namibia with four man-made waterholes, dirt roads, and game fence included. The study occurred in section A, which is a managed as a reserve, whereas section B operates as a cattle farm.

**Table 1 animals-12-03532-t001:** Independent number of visits (according to body size and species-specific) by ungulates at waterholes during the cheetah-presence and cheetah-absence treatments. In brackets, percentages of sampling sessions that had visits by the specific species, in relation to the total number of sampling sessions for the respective treatment. Small—duiker and steenbok. Medium—impala and springbok. Large—greater kudu.

Treatment	Prey Species				
	Small		Medium		Large
Cheetah present	37 (35%)		2 (2%)		66 (63%)
Chetah absent	29 (43%)		18 (27%)		21 (31%)
	Duiker	Steenbok	Impala	Springbok	Kudu
Cheetah present	3 (3%)	34 (32%)	0 (0%)	2 (2%)	66 (63%)
Cheetah absent	4 (6%)	25 (37%)	7 (10%)	11 (16%)	21 (31%)

**Table 2 animals-12-03532-t002:** Model estimates of individual visitation rate, herd visitation rate, and log-transformed dependent variable (duration of waterhole visit). Reference values are cheetah absence, large species, and season 1 (wet–hot).

	Individual Visitation Rate		Herd Visitation Rate		Duration of Stay	
Fixed effects	Estimate	Std Error	Estimate	Std Error	Estimate	Std Error
(Intercept)	−0.265	0.464	−1.115 **	0.404	1.676 ***	0.284
Cheetah presence	0.983 ^	0.527	0.816 ^	0.436	0.653 *	0.285
Body size medium	0.433	0.573	−0.152	0.503	-	-
Body size small	−0.274	0.583	0.311	0.481	0.300	0.323
Season 2 (dry–cold)	0.240	0.409	0.339	0.329	0.131	0.237
Season 3 (intermediate)	−0.214	0.415	−0.158	0.346	0.095	0.254
Cheetah presence ×	−4.927 *	1.030	−3.375 ***	0.936	—	—
Body size medium						
Cheetah presence ×	−1.255	0.775	−0.961	0.619	−0.519	0.417
Body size small						

Significance levels: 0 ‘***’ 0.001 ‘**’ 0.01 ‘*’ 0.05 ‘^’.

## Data Availability

Data used for the analysis of this investigation will be publicly posted after acceptance.

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
