# Peer review of "Rewilding Apex Predators Has Effects on Lower Trophic Levels: Cheetahs and Ungulates in a Woodland Savanna"

_animals, 2022, doi:10.3390/ani12243532_

Round 1
Reviewer 1 Report
This study is similar in aims and methodology to studies of predator reintroductions in North America and Europe but is arguably of particular interest because of the complex of predator and prey species in The studied African ecosystem. In most other studies to date there has been the reintroduction of a single top predator into an essentially predator deficient ecosystem. The results are convincing that, as expected, presence of reintroduced predators do influence prey behaviour depending on prey size increasing behaviour consistent with reduction of predator risk. The results suggest particular caution is required in reintroductions in systems where predator-prey relations have similar complexity and that close monitoring of ecosystem responses is required to establish best practice in reintroduction to ensure undesirable outcomes are minimised.
Author Response
Thank you and we really appreciate the positive assessment of our manuscript.
Reviewer 2 Report
I think this is generally a very interesting topic about the influence of apex predators on the behavior of their prey. I am a little surprised by the relatively low sample size, not in number of hours, but the sample size in total, but I understand it may be of logistical reasons. The other thing I would suggest the authors is to present some alternative hypotheses, or explanations to their findings. I believe there could be for example over different densities in the total game reserved, before and after the presence of the "experiment" of presence and absence of cheetahs. Thus, the reasons why the large ungulates stay longer at waterholes when cheetahs are present, may be due to there being much larger densities of large ungulates during this time, and/or much less densities when cheetahs are absent. I think that the authors should on overall try to present some alternative explanations to their findings. Otherwise it is an interesting study and I believe more of these studies are wanted to see the effects of apex predators influence on their prey behavior.
